# Differential Regulation of Wingless-Wnt/c-Jun N-Terminal Kinase Crosstalk via Oxidative Eustress in Primary and Metastatic Colorectal Cancer Cells

**DOI:** 10.3390/biomedicines12081816

**Published:** 2024-08-09

**Authors:** Gitana Maria Aceto, Sara Pagotto, Francesco Domenico Del Pizzo, Concetta Saoca, Federico Selvaggi, Rosa Visone, Roberto Cotellese, M’hammed Aguennouz, Rossano Lattanzio, Teresa Catalano

**Affiliations:** 1Department of Medical, Oral and Biotechnological Sciences, “G. d’Annunzio” University of Chieti-Pescara, 66100 Chieti, Italy; sara.pagotto@unich.it (S.P.); rosa.visone@unich.it (R.V.); roberto.cotellese@unich.it (R.C.); 2Center for Advanced Studies and Technology (CAST), “G. d’Annunzio” University of Chieti-Pescara, 66100 Chieti, Italy; francescodomenico.delpizzo@alumni.unich.it (F.D.D.P.); rossano.lattanzio@unich.it (R.L.); 3Department of Neurosciences, Imaging and Clinical Sciences, “G. d’Annunzio” University of Chieti-Pescara, 66100 Chieti, Italy; 4Department of Clinical and Experimental Medicine, University of Messina, 98125 Messina, Italy; concetta.saoca@unime.it (C.S.); aguenoz.mhommed@unime.it (M.A.); tcatalano@unime.it (T.C.); 5Unit of General Surgery, Ospedale Clinicizzato SS Annunziata of Chieti, 66100 Chieti, Italy; fedeselvaggi@hotmail.com; 6Villa Serena Foundation for Research, 65013 Città Sant’Angelo, Italy; 7Department of Innovative Technologies in Medicine & Dentistry, “G. d’Annunzio” University of Chieti-Pescara, 66100 Chieti, Italy

**Keywords:** Wnt/β-catenin, adenomatous polyposis coli, JNK, colorectal cancer progression, H_2_O_2_, oxidative stress, tumor microenvironment, tumor metabolism, metastasis

## Abstract

In the tumor microenvironment (TME), ROS production affects survival, progression, and therapy resistance in colorectal cancer (CRC). H_2_O_2_-mediated oxidative stress can modulate Wnt/β-catenin signaling and metabolic reprogramming of the TME. Currently, it is unclear how mild/moderate oxidative stress (eustress) modulates Wnt/β-catenin/APC and JNK signaling relationships in primary and metastatic CRC cells. In this study, we determined the effects of the H_2_O_2_ concentration inducing eustress on isogenic SW480 and SW620 cells, also in combination with JNK inhibition. We assessed cell viability, mitochondrial respiration, glycolysis, and Wnt/β-catenin/APC/JNK gene and protein expression. Primary CRC cells were more sensitive to H_2_O_2_ eustress combined with JNK inhibition, showing a reduction in viability compared to metastatic cells. JNK inhibition under eustress reduced both glycolytic and respiratory capacity in SW620 cells, indicating a greater capacity to adapt to TME. In primary CRC cells, H_2_O_2_ alone significantly increased *APC*, *LEF1*, *LRP6*, *cMYC* and *IL8* gene expression, whereas in metastatic CRC cells, this effect occurred after JNK inhibition. In metastatic but not in primary tumor cells, eustress and inhibition of JNK reduced APC, β-catenin, and pJNK protein. The results showed differential cross-regulation of Wnt/JNK in primary and metastatic tumor cells under environmental eustress conditions. Further studies would be useful to validate these findings and explore their therapeutic potential.

## 1. Introduction

Colorectal cancer (CRC) is one of the most common tumors worldwide [1]. However, CRC incidence and mortality rates show some geographical variability related to environmental and lifestyle risk factors [1,2]. Approximately 90% of CRC mortality can be attributed to metastatic disease [3,4]. The genesis of sporadic CRC is a slow process driven by complex interactions between immune surveillance of the gut microbiota and epithelial renewal capacity [2,3,5]. In most cases, CRC arises from an adenoma by accumulating somatic mutations and epigenetic alterations through interactions with the functions of the tissue microenvironment [2,6,7]. CRC usually takes years to develop; during this process, complex interactions between genetic and environmental factors allow the tumor to acquire new hallmark functions and tissue-specific growth solutions with metabolic remodeling [8,9]. CRC progression has long been attributed to the dysregulation of Wingless/It (Wnt)/β-catenin signaling [10] and its altered crosstalk with other cellular pathways necessary for the physiological renewal of the intestinal epithelium [11,12]. Furthermore, the pathogenesis and progression of CRC have been linked to the deregulation of signaling pathways induced by Wnt ligands [2,3,13,14]. The progression of metastatic CRC can be affected by factors such as the primary tumor’s traits, the tumor microenvironment, and the host immune system. In addition, treatment modalities such as surgical resection combined with chemotherapy, radiotherapy, and immunomodulatory therapy can also influence the complexity of the metastatic process [3,15,16,17]. 

Reactive oxygen species (ROS) are cellular products that are physiologically required to modulate the ability of cells to adapt to metabolic and defense demands (eustress); they can act as second messengers or signaling molecules [18]. ROS are very versatile due to their different reactivity modulating intracellular or paracellular activities [19,20]. In the gut, ROS act as multifaceted regulators of the intestinal barrier in homeostasis, infectious diseases, and intestinal inflammatory responses [21,22,23,24]. Induction of aerobic glycolysis in cancer cells can be modulated by activation of the Jun N-terminal kinases (JNK) pathway via mitochondrial ROS [25]. Furthermore, the inflammatory tumor microenvironment (TME) can intervene in this process through molecular crosstalk signals, such as stress-activated protein kinases (SAPK)/JNK and Wnt/β-catenin [26,27,28,29]. These two signals, which have been shown to be interconnected, may favor the ability of both primary and metastatic CRC cells to adapt to the variability of the tissue micro-environment in order to escape immune suppression and establish themselves in a new hospitable niche [3]. It has been reported that altered levels of ROS produced in the microenvironment of cancer cells have been linked to recurrence or relapse [30]. These changes may contribute to the transition between dormancy and metastatic spread [3,31].

Hydrogen peroxide (H_2_O_2_) is important in this context because of its ability to permeate biological membranes. It is generated in various cellular compartments as a result of mitochondrial redox processes [32]. It can be produced by membrane nicotinamide adenine dinucleotide phosphate (NADPH) oxidases in intestinal epithelial cells and by commensal microbial species in the gut [21,23]. In CRC cells, H_2_O_2_-mediated oxidative stress may modulate the canonical and noncanonical Wnt/β-catenin signaling pathways linked to the JNK pathway and the metabolic program of TME [25,33]. 

Currently, it is unclear how oxidative stress may differentially modulate Wnt/β-catenin and JNK signaling relationships in primary versus metastatic CRC, with mutant Adenomatous polyposis coli (*APC*). Considering the role of H_2_O_2_ in carcinogenesis, it is important to know how this molecule could remodulate the Wnt signaling in CRC progression and to evaluate the molecular relationship between oxidative stress and JNK/Wnt/β-catenin in primary SW480 and metastatic SW620 CRC cell models derived from the same patient. In this study, we initially determined the changes in cell viability and metabolism following H_2_O_2_-induced eustress, alone and/or in combination with JNK inhibition. We then assessed the effects of oxidative eustress on Wnt/β-catenin/JNK signaling molecules via gene and protein expression.

## 2. Materials and Methods 

### 2.1. Cell Cultures and Treatments 

The isogenic CRC progression cell models SW480 and SW620 derived from the same patient [34] were purchased from the American Type Culture Collection (ATCC, Manassas, VA, USA). SW480 cells were propagated in RPMI-1620 medium (EuroClone, Pero, MI, Italy), while the SW620 cell line was cultured in Dulbecco’s Modified Eagle Medium (DMEM) (EuroClone) in a humidified atmosphere with 5% CO_2_ at 37 °C. Both media were supplemented with 10% fetal bovine serum, 100 U/mL penicillin/streptomycin and 2 mM L-glutamine (EuroClone). For the experiments, proliferative cells at subconfluence (70%) were exposed for 3 h with differing low concentrations of H_2_O_2_ (Sigma-Aldrich, Milan, Italy) at 0.005, 0.01, 0.025, 0.05, 0.1, 0.25, 0.5, 1 mM, alone or pretreated for 60 min with the ATP competitive inhibitor of c-Jun N-terminal kinase (JNK) anthra[1,9-cd]pyrazol-6(2H)-one (SP600125) (Sigma-Aldrich, Milan, Italy) at [10 μM]. The H_2_O_2_ concentration range was chosen based on the physiological amounts of H_2_O_2_ reported as being produced by proliferating epithelial cells [19].

### 2.2. Cell Viability and Metabolic Assay by MTS

We performed the colorimetric 2-[2-methoxy-4-nitrophenyl]-3-[4-nitrophenyl]-5-[2,4-disulphophenyl]-2H-tetrazolium, monosodium salt (MTS) assay (Promega, Milan, Italy) to test cell viability and metabolic activity of SW480 and SW620 cells as primary and metastatic CRC models, respectively, through evaluation of the mitochondrial function. SW480 and SW620 cell lines were seeded in a 96-well plate at a density of 1.0 × 10^4^ cells/well; then 70% of subconfluent cells were exposed for 3 h to H_2_O_2_ at 0.005, 0.01, 0.025, 0.05, 0.1, 0.25, 0.5, 1 mM, alone or after 1 h of SP600125 pretreatment at [10 μM]. An MTS assay (Promega, Milan, Italy) was performed as previously described [33]. The absorbance was measured at 490 nm with a microplate reader by GloMax-Multi Detection System (Promega). For each experimental condition, five replicates were performed in three independent experiments.

### 2.3. Seahorse Assays

Seahorse assays were performed to determine real-time changes in the energy requirements of SW480 and SW620 living cells in response to 3-h exposure at a concentration of H_2_O_2_ inducing eustress, alone or after 1 h of SP600125 pretreatment at [10 μM]. We used the Seahorse XF Glycolytic Rate Assay kit and the Seahorse XF24 Analyzer (Agilent Technologies, Santa Clara, CA, USA) [35], a sensor cartridge system to assay oxygen consumption rate (OCR) and extracellular acidification rate (ECAR), which allow a real-time evaluation of aerobic mitochondrial respiration and anaerobic glycolysis, respectively. The protocol was set up as suggested by the manufacturer. The cells were seeded at a density of 4.0 × 10^4^ cells/well in XF24 culture plates and maintained for 24 h. Before the analysis, the culture medium was replaced, and the cells were left to equilibrate in a CO_2_-free incubator. OCR and ECAR were measured under basal conditions with SP600125 pretreatment and the successive addition of H_2_O_2_.

### 2.4. Gene Expression by Real-Time Quantitative PCR Analysis (qRT-PCR)

For gene expression assays, the cells were seeded on a 10 cm plate at a density of 3.2 × 10^4^; at subconfluence (70%), they were stressed with 0.05 mM H_2_O_2_ for 3 h alone or after JNK inhibition with SP600125. Total RNA was isolated from treated and untreated cells using EuroGold TriFast™ (EuroClone, Pero, MI, Italy), according to the manufacturer’s instructions. RNA samples were evaluated for purity and quantified by Nanodrop 1000 Spectrophotometer (Applied Biosystems, Thermo Fisher Scientific, Waltham, MA, USA). The synthesis of complementary DNA (cDNA) was obtained using the GoTaq^®^ 2 Step RT-qPCR Kit (Promega) according to the manufacturer’s instructions. The mRNA levels were measured by SYBR Green quantitative real-time PCR (qRT-PCR) analysis using StepOne™ 2.0 Real-Time PCR System (Applied Biosystems). Data were analyzed by the comparative Ct method and were indicated as 2^−ΔΔCt^ ± SD. In line with the method, the target gene’s mRNA levels were normalized by the ratio of the median value of the endogenous reference β-glucuronidase (*GUSB*) gene in treated cells vs. untreated cells. Target and reference genes were amplified in triplicate in a reaction mix of 10 μL containing 1 μL of cDNA template, 0.2 μL of primers mixture and 5 μL of GoTaq^®^ 2-Step RT-qPCR System (Promega), following the manufacturer’s instructions [36]. The sequences of paired oligonucleotides were: *fw* 5′-GCTTGATAGCTACAAATGAGGACC-3′ and *rv* 5′-CCACAAAGTTCCACATGC-3′ for *APC*; *fw* 5′-CCCATGCACCTGGTTCTACT-3′ and *rv* 5′-CCAAGCCACAGGGATACAGT-3′ for *LRP6*; *fw* 5′-GAC GAG ATG ATC CCC TTC AA -3′ and *rv* 5′-AGG GCT CCT GAG AGG TTT GT-3′ for *LEF1*; *fw* 5′-TCGACATGGAGTCCCAGGA-3′ and *rv* 5′-GGCGATTCTCTCCAGCTTCC-3′ for *JUN/AP1*; *fw* 5′-ATGACTTCCAAGCTGGCCGT-3′ and *rv* 5′-TCCTTGGCAAAACTGCACCT-3′ for *CXCL8* (IL8); fw 5′-CCTAGTGCTGCATGAGGAGA-3′ and *rv* TCTTCCTCATCTTCTTGCTCT for c*MYC*; *fw* 5′-AGCCAGTTCCTCATCAATGG-3′ and *rv* 5′-GGTAGTGGCTGGTACGGAAA-3′ for *GUSB*.

### 2.5. Protein Extraction and Western Blotting

Total proteins were isolated from two biological replicates of subconfluent cells in 10-cm plates (3.2 × 10^4^) after 0.05 mM H_2_O_2_ stress for 3 h with and without inhibition of JNK by SP600125 for 1 h. Cell samples were lysed in cold RIPA buffer supplemented with proteases and phosphatase inhibitors cocktails (Merck Group, Darmstadt, Germany). Equal amounts of total proteins were loaded on 4–20% Criterion™ TGX Stain-Free™ Protein Gel that allows rapid fluorescent detection of proteins with a stain-free imaging system (Gel Doc™ EZ System, Bio-Rad, Hercules, CA, USA). Proteins were transferred to the PVDF membrane (Merck Group, Darmstadt, Germany), blocked with 5% nonfat dry milk in phosphate-buffered saline (PBS) with 0,1% Tween 20 (Merck Group, Darmstadt, Germany), and incubated using the following primary antibodies: APC at 1:1000 dilution (PA5-30580, Thermo-Fisher Scientific, Waltham, MA, USA); β-catenin at 1:2000 dilution (Clone 14, BD Transduction Laboratories™, Becton, Dickinson and Company, Franklin Lakes, NJ, USA), E-cadherin at 1:2000 dilution (Clone NCH-38, Agilent, Santa Clara, CA, USA), Phospho-SAPK/JNK (Thr183/Tyr185) at 1:1000 dilution (Cell Signaling Technologies, Danvers, MA, USA. Detection was performed using ECL chemiluminescent substrate (Thermo-Fisher Scientific, Waltham, MA, USA) after hybridization with the specific HRP-conjugated secondary antibody (Cell Signaling Technology, Danvers, MA, USA). High-resolution digital images were acquired with a chemiluminescence and epifluorescence imaging system (Uvitec mod Alliance 9.7, Cambridge, UK). Western blot signals were quantified by ImageJ (Fiji software, 1.52 version, https://fiji.sc/), and stain-free total protein signal analysis was performed as the loading control [37].

### 2.6. Preparing Cell Blocks from SW480 and SW620 Cell Lines and Performing the Immunocytochemical Staining

For immunocytochemical (ICC) experiments, SW480 and SW620 cells were seeded at a density of 2.1 × 10^6^ per T75 flask. At sub-confluence (70%), the cells were stressed with 0.05 mM H_2_O_2_ for 3 h alone or after JNK inhibition. At least 8 × 10^6^ cells were used for each tumor cell line, as previously described in Catalano et al. 2022 [33]. Cell blocks were prepared by centrifuging cells for 10 min at 750 rpm. The resulting pellets were then placed into vials and transported to the histology lab in a 10% neutral buffered formalin solution for 1 h and processed for routine histologic sections. ICC stains were performed on 5-micrometer cell block sections. Antigen retrieval was carried out by microwave treatment at 750 W for 10 min using a citrate buffer (pH 6.0) before the ICC staining was performed with the following primary antibodies: anti-APC at 1:100 dilution (PA530580; Thermo-Fisher Scientific, Waltham, MA, USA), β-catenin at 1:3000 dilution (610154; BD Transduction Laboratories™, Becton, Dickinson and Company, Franklin Lakes, NJ, USA) and E-cadherin at 1:50 dilution (610181-82; BD Transduction Laboratories™). 3,3′-diaminobenzidine (DAB) was used as chromogen. Slides were then counterstained with hematoxylin, cleared, and mounted. In negative control sections, the specific primary antibodies were replaced with nonimmune serum or isotype-matched immunoglobulins. The expression of molecular markers was assessed based on the presence of distinct specific staining in tumor cells, expressed as a percentage of stained cells, and the intensity of staining.

### 2.7. Statistical Analysis

This research was primarily conducted in vitro using two validated models of colon cancer progression derived from the same patient, SW480 from primary CRC and SW620 from lymph node metastasis [34]. All measurements were made after at least three independent experiments, and for each set of data, a representative value of all experiments plus standard deviation is shown. The results were subjected to a *t*-test or one-way analysis of variance (ANOVA) as appropriate. All *p*-values are two-tailed, and a *p*-value of less than 0.05 was considered significant. All analyses were performed using SPSS software version 20 (IBM Corp., Armonk, NY, USA).

## 3. Results

### 3.1. Cell Viability after Oxidative Eustress Induced by H_2_O_2_ with and without JNK Inhibition

In order to determine the changes in cell viability and metabolic activity by H_2_O_2_-induced eustress, alone and/or combined with JNK inhibition, we performed an MTS assay, as described in Methods. Primary SW480 cell line and its lymph node metastatic variant SW620 were exposed for 3 h to low concentrations of H_2_O_2_ (0.005, 0.01, 0.025, 0.05, 0.1, 0.25, 0.5, 1 mM), alone or after 1 h of SP600125 pretreatment (Figure 1). For each experiment, five wells were analyzed as biological replicates for each treatment. Low concentrations of H_2_O_2_ alone caused different effects in isogenic SW480 and SW620 cell lines (Figure 1a,b). Compared to untreated cells, we observed reduced viability and mitochondrial activity in primary CRC cells, persisting at higher H_2_O_2_ concentrations 0.1, 0.25, 0.5, and 1 mM, in contrast to metastatic SW620 cells (Figure 1a,b). We then noted that the treatment with H_2_O_2_ [0.05 mM] induced no significant change in cell viability of either primary or metastatic CRC cells (Figure 1a,b). The identification of this eustress concentration from H_2_O_2_ allowed us to evaluate its effects in combination with JNK inhibition. Primary CRC cells (SW480) appeared more sensitive to the action of H_2_O_2_ [0.05 mM] combined with JNK inhibition, showing a reduction in cell viability, whereas in SW620 metastatic cells this did not occur. Indeed, the pretreatment with the inhibitor SP600125 and subsequent exposure to H_2_O_2_ [0.05mM] for 3 h reduced the viability and mitochondrial activity in the primary CRC model compared to control and H_2_O_2_ exposure alone, while the SW620 cells from metastasis showed no significant effect at the same conditions (Figure 1c,d). This demonstrates a different mitochondrial adaptation to JNK inhibition and oxidative eustress in primary CRC cells 480 compared to its matched lymph node metastatic variant.

### 3.2. Oxygen Consumption Rate and Extracellular Acidification Rate Measurements

We evaluated differences in key metabolic parameters of mitochondrial function in response to H_2_O_2_ eustress and JNK inhibition in primary (SW480) and metastatic (SW620) CRC cells from the same patient. The measurements of oxygen concentration and proton flux in the cell supernatant by Seahorse assay are converted into oxygen consumption rate (OCR) and extracellular acidification rate (ECAR) values, allowing the direct quantification of mitochondrial respiration and glycolysis in SW480 and SW620 cells (Figure 2 and Figure 3). Tumor cells in active replication must be able to reproduce their nucleic acids. The precursors essential for this synthesis are obtained from intermediates of glycolysis via a metabolic pathway known as the pentose shunt. OCR and ECAR measurements in living cells by Seahorse XF24 show that primary cancer cells SW480 and metastatic cancer cell SW620 have their own individual metabolic features depending on H_2_O_2_ exposure and JNK inhibition. Inhibition of JNK in primary CRC cells induced an increase in mitochondrial respiration and glycolytic capacity that was reversed by the addition of H_2_O_2_ [0.05 mM] (Figure 2a–d). In metastatic SW620 cells, the treatment with H_2_O_2_ alone or with SP600125 alone did not induce any modulation compared to the untreated control. Instead, the combination of the JNK inhibitor with H_2_O_2_ [0.05 mM] reduced both glycolytic and respiratory capacity (Figure 3a–d), thus denoting a greater capacity for adaptation of metastatic cells to the microenvironment and nutrient bioavailability.

### 3.3. Gene Expression after Oxidative Stress Induced by H_2_O_2_ and JNK Inhibition

To investigate whether JNK inhibition and H_2_O_2_ eustress exposure induced different gene expression levels in Wnt signaling, *cMYC*, and *IL8* in both SW480 and SW620 CRC cell lines, we performed real time-qPCR (Figure 4a,b). Combined JNK inhibitor SP600125 pre-incubation for 1 h and subsequent H_2_O_2_ [0.05 mM] exposure for 3 h resulted in overexpression of *APC*, *LRP6*, *LEF1*, *JUN/AP1*, *cMYC*, and *IL8* in metastatic SW620 cells. Conversely, the primary SW480 cell line showed a higher expression of the same genes, compared to the control, only after treatment with H_2_O_2_ [0.05 mM] alone, excluding *JUN/AP1*, whose expression increased in the primary CRC cells pretreated with SP600125 followed by H_2_O_2_ [0.05 mM] exposure.

### 3.4. Protein Expression after H_2_O_2_ Eustress

The effect induced by H_2_O_2_ eustress in proliferative SW480 and SW620 cells was also assessed on APC, β-catenin, and E-cadherin protein expression. Western blotting analyses revealed an increase of the full-length APC protein (310 kDa band) in the primary SW480 cell line after H_2_O_2_ treatment. In comparison, in metastatic SW620 cells, the full-length APC protein (310 kDa band) was decreased by H_2_O_2_ and JNK inhibition alone and combined when compared to the analogous untreated cells. Both cells carry an *APC* mutation that produces a truncated APC protein of 147 KDa. This isoform is present in the CRC proliferating cells and showed a slight reduction after treatments. In contrast, the APC low molecular weight (LMW) band (approximately 70 kDa) was reduced by the JNK inhibitor alone and its combination with H_2_O_2_. This effect appeared more pronounced in SW620 cells from metastasis than in SW480 from the primary tumor (Figure 5). H_2_O_2_ eustress and JNK inhibition did not induce relevant changes in the β-catenin expression in SW480 cells, whereas it decreased in SW620 cells (Figure 5).

### 3.5. Expression of Wnt Pathway Components in SW480 and SW620 Cell Block Sections

Changes in the protein expression of APC, β-catenin, and E-cadherin induced by oxidative stress and JNK inhibitor SP600125 were detected by ICC. In the SW480 cell line, H_2_O_2_ [0.05 mM] increased cytoplasmic APC expression from 5% to 80% of tumor cells. The inhibitor SP600125, alone or combined with H_2_O_2,_ resulted in APC protein expression at 5% and 10%, respectively. Instead, the levels of β-catenin and E-cadherin expression remained unchanged after treatments compared to untreated cells with levels of expression that remained at 5–10% and below 5%, respectively (Figure 6). In the SW620 metastatic cell line, H_2_O_2_ [0.05 mM] reduced the cytoplasmic APC expression from 40% to 10% compared to the control. Treatment with the JNK inhibitor SP600125, alone or combined with H_2_O_2_, decreased APC expression to less than 5% of tumor cells. Additionally, treatments with H_2_O_2_ alone and H_2_O_2_ combined with SP600125 caused β-catenin membrane expression in 20% of cancer cells, compared to 90% and 70% in untreated and SP600125-treated metastatic cell lines, respectively. The levels of E-cadherin expression remained unchanged after treatments, compared to untreated cells, with expression levels consistently below 5% (Figure 7).

## 4. Discussion

In this study, we determined the effects of the H_2_O_2_ concentration inducing eustress by assessing cell viability and mitochondrial activity in isogenic SW480 and SW620 cells. We then evaluated the H_2_O_2_ effects in combination with JNK inhibition. Our results showed that the primary CRC cells were more sensitive to the effects of H_2_O_2_ [0.05mM] eustress in combination with JNK inhibition, showing a reduction in their viability. This effect was not seen in metastatic cells SW620. In addition, JNK inhibition combined with oxidative eustress reduced both glycolytic and respiratory capacity in the SW620 CRC cell line, indicating a greater ability of metastatic cells to adapt to the microenvironment and nutrient bioavailability. These data confirmed the involvement of JNK pathways in the control of the Warburg effect in cancer progression [25]. Mitochondrial ROS, particularly H_2_O_2_, are key participants in several signaling pathways underlying the control of cell proliferation and differentiation. Cancer cells show elevated ROS levels during all stages of malignancy [38]. Cells can adapt to oxidative stress by genetic reprogramming in the longer term and metabolic reprogramming in the short term [39]. Normal metabolism and steady-state functions require low levels of oxidants, termed “oxidative eustress”. A supraphysiological challenge with oxidants or their inadequate detoxification/inactivation is termed “oxidative distress” [40]. In the gut, H_2_O_2_ production is required to regulate the immune functions of the intestinal barrier and the homeostasis of cell renewal [21,22,23,24]. Indeed, some observational/mechanistic studies have demonstrated the production of H_2_O_2_ in the colon epithelium by microbial products, innate immune defenses, and exposure to environmental organic pollutants [21,41,42,43].

Generated by cellular demands in response to physiological stimuli of metabolic adaptation and environmental stress, H_2_O_2_-mediated eustress is involved in the inflammatory response and aerobic glycolysis responsible for lactate production by promoting mitochondrial oxidative metabolism, driving tumor cell growth and chemo-resistance [30,44,45]. Considering the role played by H_2_O_2_ in carcinogenesis, it is important to know how H_2_O_2_ might remodulate the Wnt signal in CRC cells since not all tumors respond effectively to therapy and cells can adapt to drug-induced oxidative stress. Balanced levels of ROS mediate chemotherapy resistance and allow tumor cells to survive during treatment, developing stemness and cancer-initiating capacity [38]. Moreover, dormant tumor cells resist oxidative stress as a result of their inability to consume glucose as an energy source. Dormant CRC cells persist after anticancer treatments or are responsible for drug resistance and tumor recurrence [3,39]. Therefore, H_2_O_2_ modulation within the TME may be an attractive therapeutic approach, as cancer cells have an altered redox balance compared to their normal counterparts [46,47,48,49,50,51]. Some therapies increase ROS production and redox stress as side effects or, together with the effects of chemotherapy, may induce the activation of the immune response and apoptosis in tumor cells [33,49,50,51,52,53,54]. Conversely, high levels of ROS in the TME inhibit T-cell cytotoxicity, allowing tumor invasion and treatment resistance during cancer progression. Therefore, reducing ROS levels within the TME may reduce the immunosuppressive activity of regulatory T-cells (Tregs). Lower levels of ROS in the tumor environment increase the efficacy of programmed cell death protein 1 (PD-1) blockade immunotherapy [55].

Many studies have reported that JNK is involved in CRC progression [26,56]. JNK is a member of the MAPK family of proteins that is predominantly activated by stress stimuli. Selective inhibition of JNK has recently been proposed as a target in the context of cancer therapies [57,58]. JNK regulates multiple transcriptional activities and contributes to tumor-promoting processes, ranging from cell proliferation to apoptosis, inflammation, metastasis, and angiogenesis in TME [59]. The prosurvival role of JNK is mainly mediated by crosstalk with other pathways, including the Wnt signaling [33,59]. We have previously observed that, after cellular starvation, acute oxidative distress induces a sharp increase in the expression of the truncated isoform of APC, plausibly linked to its apoptotic activities [33]. Moreover, the gene expression of *APC* correlated with that of the JUN/AP1 transcription factor [33]. APC is involved in several physiological processes that regulate the homeostasis of colon epithelium renewal, including cell cycle progression, migration, differentiation and apoptosis. *APC* mutations were initially identified in a predisposition hereditary syndrome to CRC, known as familial adenomatous polyposis (FAP) of the colon [60], although somatic inactivation of this gene is found in approximately 85% of all colorectal adenomas and carcinomas as an early event in colon cancer progression [61,62]. Recent evidence has shown that APC can regulate the response of tumor cells to chemotherapy by modulating epithelial and TME signals [63]. The results of our study demonstrate that H_2_O_2_ oxidative eustress and JNK inhibition can differentially regulate Wnt/β-catenin signaling and APC expression in primary and metastatic CRC cells, suggesting a role in the molecular relationship between JNK and Wnt/β-catenin functions in response to the oxidation state of TME (Figure 4, Figure 5 and Figure 7). The APC protein is an active caretaker of the Wnt/β-catenin signal, which ensures the homeostasis of intestinal mucosal renewal. Furthermore, the activity of Wnt pathway is supported by the metabolic gradient from glycolysis to mitochondrial oxidative phosphorylation along the crypt-villus axis. In this context, APC-dependent physiological apoptosis requires the production of ROS by the mitochondrial respiratory chain [64].

Cancer cells display the Warburg-like metabolic phenotype, characterized by high glucose uptake and conversion to lactic acid under aerobic conditions to enhance the availability of substrates. The MAPK/JNK pathways can be considered as key regulators of the Warburg effect during tumor progression [25]. Mitochondrial and cellular H_2_O_2_ may induce dual functions by activating JNK isoform phosphorylation through negative or positive regulation of aerobic glycolysis. Reprogramming of glucose metabolism by tumor cells results from mutations in oncogenes and tumor suppressor genes [25,65]. Using MTS and extracellular flux analysis, we demonstrated that CRC cells readapted mitochondrial respiration and glycolysis under H_2_O_2_ eustress stimulation and JNK inhibition. The mitochondrial oxidative phosphorylation system produces a large amount of ATP, giving tumor cells a greater ability to proliferate [66]. It has been reported that phosphorylation of the ERK1/2 signaling pathway is reduced in the early stages of colon tumorigenesis, whereas it increases in advanced metastatic CRC. This regulation of the kinase activity in CRC appears to be related to H_2_O_2_. It has also been suggested that ERK2 may have a cell pro-growth role, whereas ERK1 plays a regulatory role in colon tumorigenesis [67]. Under physiological conditions, intracellular H_2_O_2_ can be efficiently converted into a hydroxyl radical, leading to an overall increase in 8-oxoguanine (8-oxoG). This molecule is involved in cellular metabolic responses and can induce DNA lesions in proliferating cells. Oxidative DNA damage is rapidly repaired by the Base Excision Repair system, restoring the redox balance. Furthermore, the elevated levels of 8-oxoG per se may not increase cell death [68].

A deeper insight into the mechanisms of oxidative stress-induced metabolic reprogramming in TME may help to understand the pathophysiological association between metastatic tumor cells and recipient tissues. Our results showed that, in primary and metastatic CRC cells, the oxidative eustress environment differentially modulates APC and β-catenin and mitochondrial oxygen consumption and that JUN signaling may interfere with this response, thereby increasing the Warburg effect. We have shown that metabolic reprogramming in CRC is not only related to the Wnt/β-catenin and *APC* mutation status of cancer cells but is also closely associated with TME eustress. We suggest that H_2_O_2-_mediated oxidative eustress may also modulate the canonical and noncanonical Wnt/β-catenin signaling pathways in CRC cells, which are linked to the JNK pathway and the metabolic reprogramming of the TME. Further in vivo studies and clinical investigations would be beneficial to validate these findings and explore their therapeutic potential.

## 5. Conclusions

This study suggests that H_2_O_2_-mediated oxidative eustress in TME differentially modulates the crosstalk between Wnt/β-catenin and JNK signaling in primary CRC versus metastatic CRC. Understanding how primary and metastatic CRC cells adapt to an H_2_O_2_-producing intestinal micro-environment might provide new insights for therapeutic interventions [59]. The direct clinical relevance of our findings to disease outcomes and treatment strategies for CRC may need to be further explored and validated in clinical trials or patient studies. Indeed, the pathophysiology of CRC progression is very complex, and future studies will have to take into account not only the tissue micro-environment but also the patient’s systemic macro-environment. Translational studies and integrated expertise will, therefore, be needed to use the patient’s biological data for a personalized therapeutic approach to CRC and patient lifestyle care.

## Figures and Tables

**Figure 1 biomedicines-12-01816-f001:**
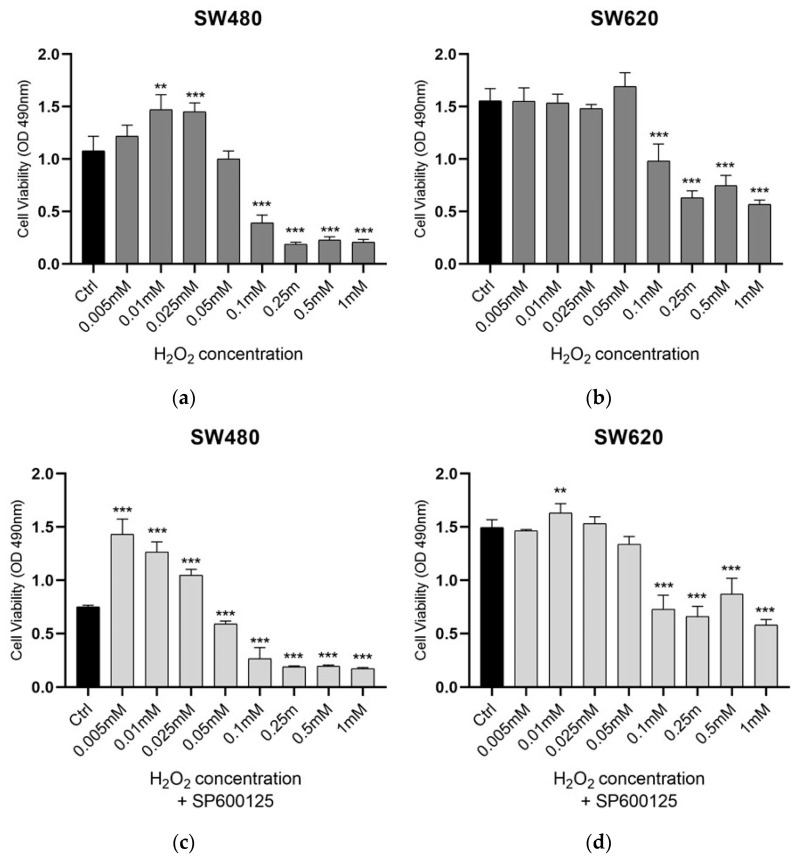
Cell-line viability under H_2_O_2_ low concentrations condition. Cell viability was assayed using MTS Dye after 3 h of treatment with H_2_O_2_ on proliferative cells: (**a**) SW480, (**b**) SW620 were treated with H_2_O_2_ at concentrations of 0.005, 0.01, 0.25, 0.05, 0.1, 0.25, 0.5, 1 mM and timing of 3 h. Cell viability values were calculated as means ± SD and compared to untreated proliferative cells (Ctrl). After 3 h of treatment, the highest concentration of H_2_O_2_ that did not induce significant changes in the mitochondrial activity of either cell turned out to be 0.05 mM (**a**,**b**). Thus, JNK inhibition was performed for one hour and subsequent exposure to H_2_O_2_ [0.05 mM] for 3 h (**c**,**d**). A reduction in mitochondrial viability and activity was observed in primary CRC cells SW480 compared to the untreated control (**c**), but also compared to H_2_O_2_ [0.05mM] exposure alone (**a**). This shows that the different mitochondrial adaptations of primary tumor CRC cells compared to its metastatic variant might depend on the JNK pathway. ** *p* < 0.01, *** *p* < 0.001 treated vs. untreated cells.

**Figure 2 biomedicines-12-01816-f002:**
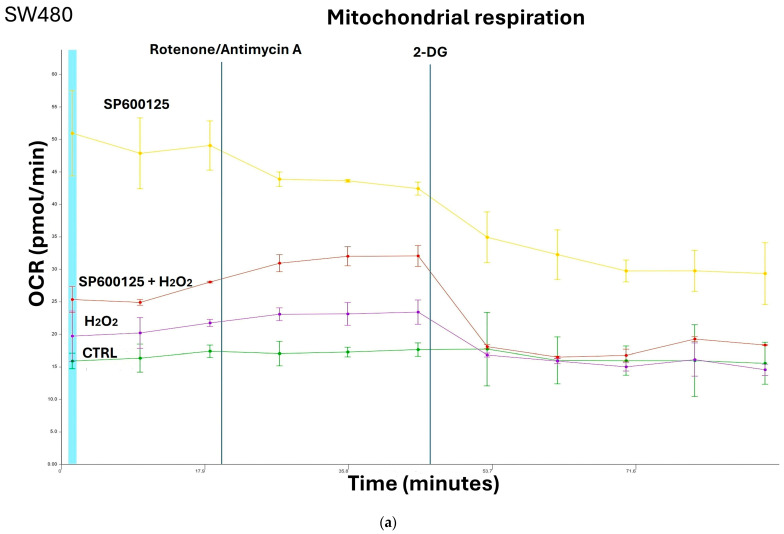
Primary cancer cells SW480 show their own individual metabolic features depending on JNK inhibition and H_2_O_2_ [0.05 mM] exposure. (**a**) OCR was investigated to assess mitochondrial respiration in SW480 cells by Seahorse analyzer. Compared to untreated cells (CTRL), pretreatment with SP600125 for 1 h increased OCR whereas, in the presence of the glycolysis inhibitor 2-DG, OCR decreased; (**b**) Energy map shows the changes in mitochondrial respiration and glycolysis of SW480 cells from basal conditions to SP600125 pretreatment for 1 h and H_2_O_2_ [0.05 mM] treatment for 3 h; (**c**) The glycolytic capacity in SW480 cells was measured by analyzing ECAR using the Seahorse analyzer. Compared to CTRL, pretreatment with SP600125 for 1 h increased ECAR values, which were instead significantly reduced in the presence of 2-DG; (**d**) Association of JNK inhibitor and H_2_O_2_ increased the oxygen consumption compared to treatment of cells with hydrogen peroxide alone. Oxygen consumption rate (OCR); extracellular acidification rate (ECAR); 2-deoxy-glucose (2-DG); hydrogen peroxide (H_2_O_2_); control (CTRL).

**Figure 3 biomedicines-12-01816-f003:**
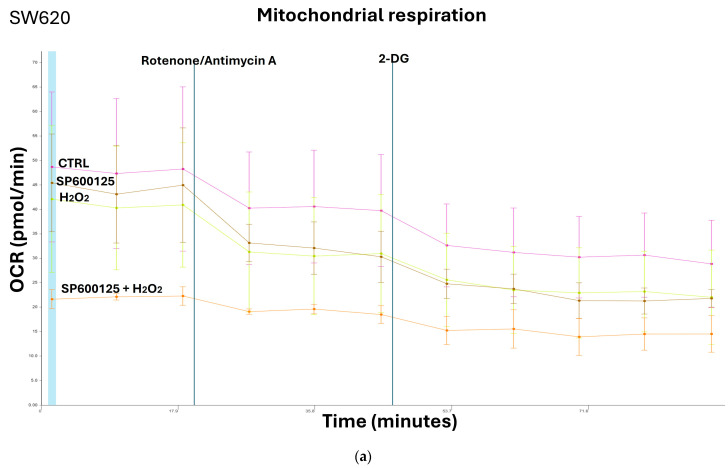
Metastatic cancer cells SW620 show metabolic features based on exposure to JNK inhibition and H_2_O_2_ [0.05 mM]. (**a**) Mitochondrial respiration in SW620 cells, evaluated as OCR by Seahorse analyzer assay, decreased significantly after pretreatment with SP600125 and subsequent addition of H_2_O_2_ [0.05 mM], compared to untreated cells (CTRL); (**b**) Energy map showed no modulation of mitochondrial respiration and glycolysis in the presence of SP600125 alone and H_2_O_2_ [0.05 mM] alone; (**c**) Glycolytic capacity, measured as ECAR, revealed no significant increase after SP600125 pretreatment in SW620 cell line, whereas H_2_O_2_ treatment alone or the association of SP600125 and H_2_O_2_ induced ECAR decrease compared to what occurred in CTRL; (**d**) Combination of JNK inhibitor and H_2_O_2_ reduced the oxygen consumption in metastatic SW620 cells compared to the same ones after incubation with SP600125 alone or hydrogen peroxide alone. Oxygen consumption rate (OCR); extracellular acidification rate (ECAR); 2-deoxy-glucose (2-DG); hydrogen peroxide (H_2_O_2_); control (CTRL).

**Figure 4 biomedicines-12-01816-f004:**
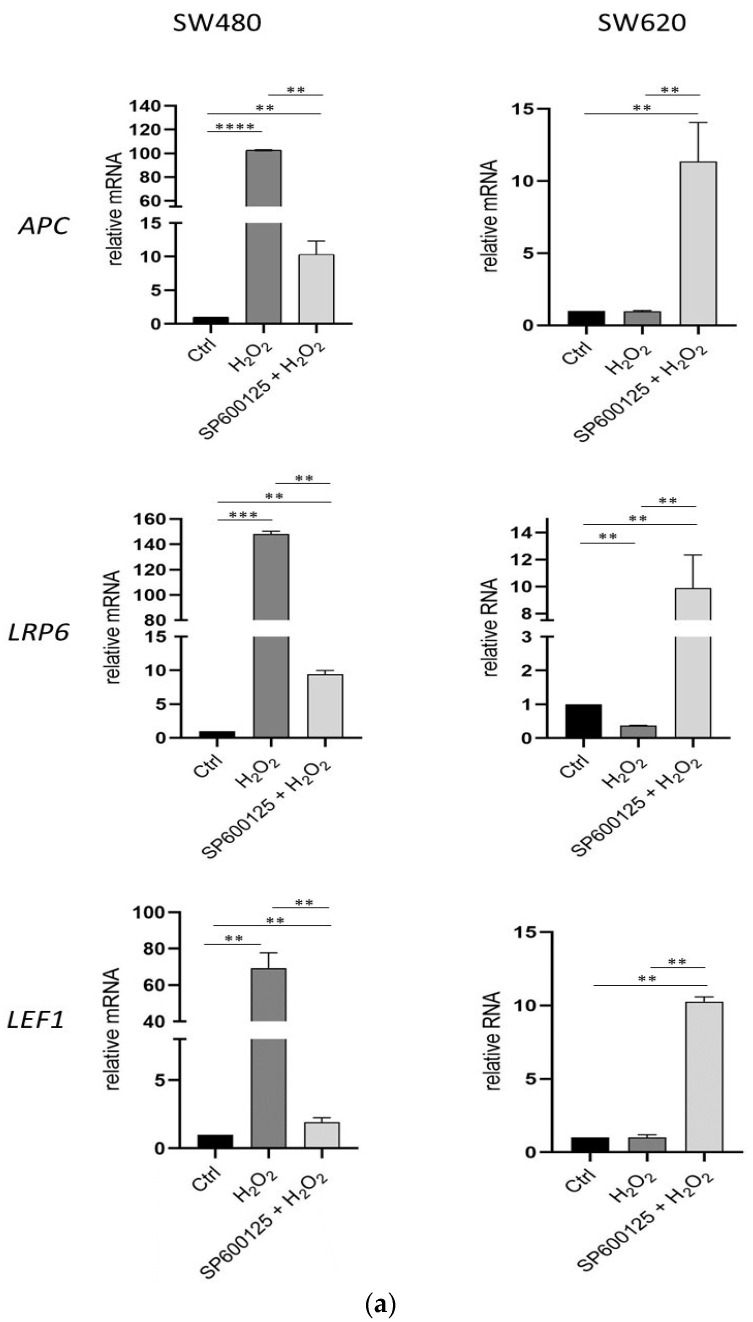
Expression of canonical and noncanonical Wnt signaling genes after 3 h of oxidative stress induced by H_2_O_2_ [0.05 mM] and JNK inhibition in primary SW480 and metastatic SW620 cells. Gene expression was analyzed by real-time-qPCR. The histograms represent data normalized with the *GUSB* gene. The results showed the average ± SD of three independent experiments. * *p* < 0.05, ** *p* < 0.01, *** *p* < 0.001, **** *p* < 0.0001 treated vs. untreated cells. (**a**): *APC*, adenomatous polyposis coli; *LRP6*, low-density lipoprotein receptor-related protein 6; *TCF/LEF*, T-cell factor/lymphoid enhancer factor. (**b**): *c-Jun/AP1*, Jun proto-oncogene/AP-1 transcription factor subunit; *cMYC*, MYC proto-oncogene; *IL8*, CXCL8, interleukin 8.

**Figure 5 biomedicines-12-01816-f005:**
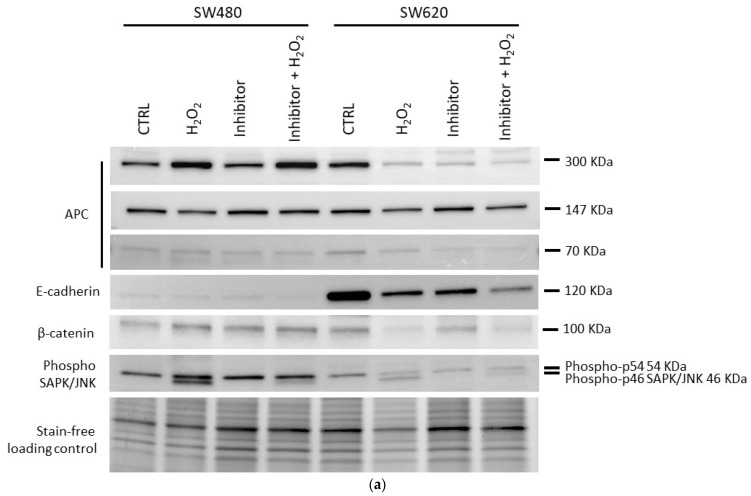
Protein expression of APC, β-catenin, E-cadherin, and phospho-SAPK/JN, in SW480 vs. SW620 cells. The cells were treated with H_2_O_2_ [0.05 mM] and JNK inhibitor SP600125. (**a**) Protein expression detected by Western blotting analysis of representative experiment from two biological replicates. The average expression levels of panel (**b**) were determined by densitometric analysis and calculated in relation to the stain-free loading control. kDa: Kilodalton as protein molecular weight unit.

**Figure 6 biomedicines-12-01816-f006:**
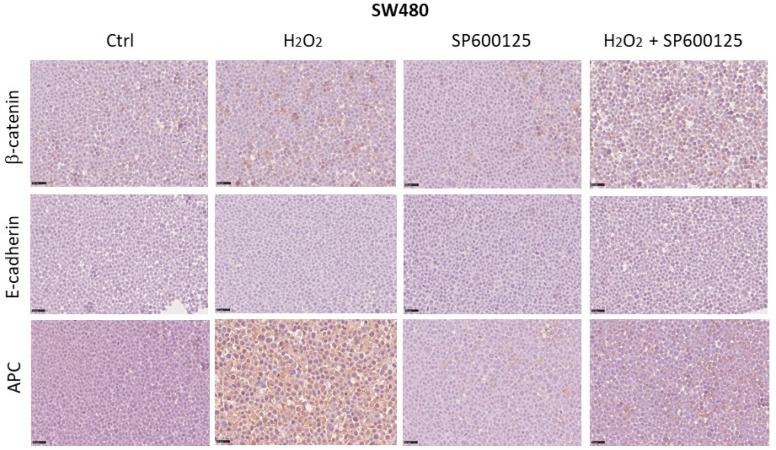
Changes in protein expression of APC, β-catenin, and E-cadherin induced by oxidative stress and JNK inhibition were detected by ICC in the SW480 cell line. Scale bars = 40 μm.

**Figure 7 biomedicines-12-01816-f007:**
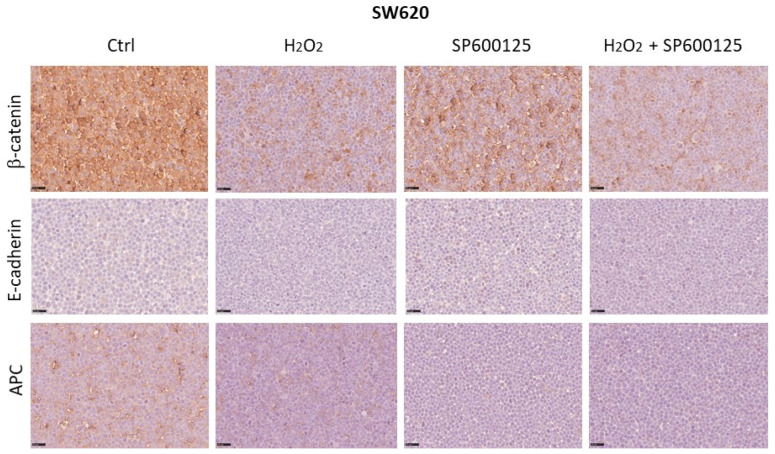
Changes in protein expression of APC, β-catenin, and E-cadherin induced by oxidative stress and JNK inhibition were detected using ICC in the SW620 cell line. Scale bars = 40 μm.

## Data Availability

Data is contained within the article.

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
