# Peer review of "Differential Regulation of Wingless-Wnt/c-Jun N-Terminal Kinase Crosstalk via Oxidative Eustress in Primary and Metastatic Colorectal Cancer Cells"

_biomedicines, 2024, doi:10.3390/biomedicines12081816_

Round 1

Reviewer 1 Report

Comments and Suggestions for Authors

Aceto et al. presented data on different Wingless-WNT/C-JUN N-Terminal 2 Kinase pathway regulations in Oxidative Eustress in Primary and Metastatic Colorectal Cancer Cells.

Major comments:

The paper's general idea is to present the different status of Wingless-WNT/C-JUN pathway proteins under H2)2-induced oxidative stress in primary SW480 and metastatic SW620 colon cancer cells. Even if the WNT-pathway part is novel, data showing viability of both cell lines after H202 treatment, WB results, as well as ICC results have already been published in the previous paper of the same group (Oxidative Distress Induces Wnt/β-Catenin Pathway Modulation in Colorectal Cancer Cells: Perspectives on APC Retained Functions, Cancers 2021 Dec; 13(23): 6045. ). Thus, I have some doubts concerning ethical issues and the novelty of the data.

  1. It is essential to consider using an alternative ROS-inducer instead of H2O2 to assess whether the observed effects correspond to ROS in general or specific effects of H2O2. Moreover, using an antioxidant compound to exclude other cellular effects of H2O2 treatment is crucial. This approach could provide more convincing and valid results, as the presence of antioxidants could prevent the cytotoxic action of H2O2, ensuring the integrity of the research.
  2. Line 210 - the Authors claim that: Three-hour treatment with H2O2 at low concentrations induces an opposite response be-210 tween cells of primary and metastatic tumours especially at [0.05mM]. I disagree with this conclusion. The H2O2 treatment in both cell lines appeared to be cytotoxic and reduced cell viability. The only difference is the sensitivity to H2O2 action, but still, in both cell lines, 0,1 mM was cytotoxic. Primary cells were more susceptible to H2)2 action, and 0,05 mM reduced cell viability, whereas in metastatic cells, this concentration was not effective. Still, the overall trend is similar, and we can't say that H202 exerts the opposite effect in metastatic cells.

Minor issue:

  1. Please update Figure 1 from Exel to more professional graphs like Figure 4.
  2. Please add the statistical difference in Figure 1 (0,05 mM)
  3. Please change comes to dots (0.01)
  4. Please add a description of how viability was calculated on the Y axis. Cell viability [......... ratio?/]
  5. Please add Ab dilutions in the WB and ICC method descriptions.
  6. Please, enlarge Figures 2 and 3
  7. Please, add the quantification analysis to Figure 7
  8. Please, add densitometric analysis to Figure 5. The protein loading is not equal; thus, bar charts are needed.
  9. Please correct the abstract and explain all abreviations that appear firs time in the text (RPS, JNK, APC)

Comments on the Quality of English Language

 Minor editing of English language required

Author Response

1° Review Report (Round 1)

Major comments:

The paper's general idea is to present the different status of Wingless-WNT/C-JUN pathway proteins under H2O2-induced oxidative stress in primary SW480 and metastatic SW620 colon cancer cells. Even if the WNT-pathway part is novel, data showing viability of both cell lines after H2O2 treatment, WB results, as well as ICC results have already been published in the previous paper of the same group (Oxidative Distress Induces Wnt/β-Catenin Pathway Modulation in Colorectal Cancer Cells: Perspectives on APC Retained Functions, Cancers 2021 Dec; 13(23): 6045 ). Thus, I have some doubts concerning ethical issues and the novelty of the data

  1. It is essential to consider using an alternative ROS-inducer instead of H2O2 to assess whether the observed effects correspond to ROS in general or specific effects of H2O2. Moreover, using an antioxidant compound to exclude other cellular effects of H2O2 treatment is crucial. This approach could provide more convincing and valid results, as the presence of antioxidants could prevent the cytotoxic action of H2O2, ensuring the integrity of the research.
  2. Line 210 - the Authors claim that: Three-hour treatment with H2O2 at low concentrations induces an opposite response be-210 tween cells of primary and metastatic tumours especially at [0.05mM]. I disagree with this conclusion. The H2O2 treatment in both cell lines appeared to be cytotoxic and reduced cell viability. The only difference is the sensitivity to H2O2 action, but still, in both cell lines, 0,1 mM was cytotoxic. Primary cells were more susceptible to H2O2 action, and 0,05 mM reduced cell viability, whereas in metastatic cells, this concentration was not effective. Still, the overall trend is similar, and we can't say that H2O2 exerts the opposite effect in metastatic cells.

Minor issue:

  1. Please update Figure 1 from Exel to more professional graphs like Figure 4.
  2. Please add the statistical difference in Figure 1 (0,05 mM)
  3. Please change comes to dots (0.01)
  4. Please add a description of how viability was calculated on the Y axis. Cell viability [......... ratio?/]
  5. Please add Ab dilutions in the WB and ICC method descriptions.
  6. Please, enlarge Figures 2 and 3
  7. Please, add the quantification analysis to Figure 7
  8. Please, add densitometric analysis to Figure 5. The protein loading is not equal; thus, bar charts are needed.
  9. Please correct the abstract and explain all abreviations that appear firs time in the text (RPS, JNK, APC)

Autors rebuttal.

Dear reviewer,

Thank you for your comments, which provide us with the opportunity to improve what was unclear and incomprehensible in the first draft of this paper. Please find below a detailed point-by-point response to all comments (reviewers’ comments in black, our replies in blue).

As you say, the work is original and above all unique. The previous article, published in 2021, was conducted on 3 CRC cell lines of which 2 were used again in this new study. But with different purposes.

In the first study the CRC models HCT116 (MSI) with APC wt and SW480 (MSS) with mutated APC, allowed us to show that MSI and MSS tumours, under dormancy conditions, have different behaviours in response to acute oxidative distress by remodelling the canonical and non-canonical WNT/b-catenin signal pathway. Most importantly, they revealed a novel behaviour for the APC protein, namely the expression of a protein isoform involved in the apoptotic response. Furthermore, gene expression data showed that APC gene expression correlated with that of the JUN/AP1 transcription factor. In 2021, experiments were conducted with high amounts of H2O2 (DISTRESS) for a very short time (15 minutes) on starved CRC cells.

In this new study we presented, we wanted to delve deeper into the interactions between Wnt/b-catenin and JNK in proliferative primary cellular models of CRC (SW480) and related metastases (SW620) bearing somatic APC mutation. These two cells are the unique isogenic models currently available derived from the same patient and responding to the characteristics of a sporadic CRC [Hewittet al., 2000]. Taking all of this into consideration, we simulated, in the TME, oxidative changes that did not a priori cause their death, to evaluate the metabolic adaptation capacity of CRC cells and the effect on Wnt/b-catenin/APC signaling. The MTS experiments allowed us to identify what this environmental condition of oxidative EUSTRESS was, and then apply it again by inhibiting JNK. Normal metabolism and steady-state functions require low levels of oxidants, termed “oxidative eustress”. A supraphysiological challenge with oxidants or their inadequate detoxification/inactivation is termed “oxidative distress” [40]. In the gut, H2O2 production is required to regulate the immune functions of the intestinal barrier and the homeostasis of cell renewal [21-24]. Indeed, some observational/mechanistic studies have detected the production of H2O2 in the colon epithelium by microbial products, innate immune defence and environmental organic pollutants exposure [41-44].

  1. For this study, the use of H2O2 was essential because it is a more stable ROS than other cellular products; furthermore, it is a diffusible second messenger, normally produced by the colon epithelial cells themselves when stimulated by metabolic and infectious agents in their microenvironment. The use of antioxidant substances could have confounded the detection of the effects of H2O2 eustress on Wnt/b-catenin and APC signaling and their relationships with JNK.
  2. We thank the reviewer for these comments which allow us to correct what we have erroneously written in the text.

In order to determine the changes in cell viability and metabolic activity by H2O2-induced eustress, alone and/or combined with JNK inhibition, we performed MTS assay. as described in Methods. Primary SW480 cell line and its lymph node metastatic variant SW620, were exposed for three hours to low concentrations of H2O2 (0.005, 0.01, 0.025, 0.05, 0.1, 0.25, 0.5, 1 mM), alone or after 1 hour of SP600125 pretreatment (Figure 1). For each experiment, five wells were analyzed as biological replicates for each treatment.  Low concentrations of H2O2 alone caused different effects in isogenic SW480 and SW620 cell lines (Figure 1. a, b). Compared to untreated cells, we observed reduced viability and mitochondrial activity in primary CRC cells, persisting at higher H2O2 concentrations 0.1, 0.25, 0.5, 1 mM, in contrast to metastatic SW620 cells (Figure 1. a, b). We then noted that the treatment with H2O2 [0.05 mM] induced no significant change in cell viability of either primary or metastatic CRC cells (Figure 1. a, b). The identification of this eustress concentration from H2O2 allowed us to evaluate its effects in combination with JNK inhibition. Primary CRC cells (SW480) appeared more sensi-tive to the action of H2O2 [0.05 mM] combined with JNK inhibition, showing a reduc-tion in cell viability, whereas in SW620 metastatic cells this did not occur. Indeed, the pre-treatment with the inhibitor SP600125 and subsequent exposure to H2O2 [0.05mM] for 3 hours reduced the viability and mitochondrial activity in the primary CRC model compared to control and H2O2 exposure alone, while the SW620 cells from metastasis showed no significant effect at the same conditions (Figure 1. c, d). This demonstrates a different mitochondrial adaptation to JNK inhibition and oxidative eustress in primary CRC cells 480 compared to its matched lymph node metastatic variant. These data confirmed the involvement of JNK pathways in the control of the Warburg effect in cancer progression [Papa et al 2018].

Minor issue:

  1. We improved the graphics in Figure 1 using the “graphpad” software.
  2. We double-checked all experiments performed and entered statistical significance, where it was present.
  3. We changed comes to dots (0.01)
  4. We added the description of how viability was calculated on the Y axis.
  5. We added Ab dilutions in the WB and ICC method descriptions.
  6. We enlarged Figures 2 and 3
  7. We added the quantification analysis to the ICC results corresponding to Figures 6 and 7.
  8. We added the densitometry analysis to Figure 5.
  9. We corrected the abstract and explained all abbreviations that appear first time in the text.

Reviewer 2 Report

Comments and Suggestions for Authors

These findings shed light on the intricate interplay between oxidative stress, signaling pathways, and metabolic adaptations in CRC cells, offering potential avenues for targeted therapeutic approaches. The study highlights that metabolic reprogramming in CRC is not only associated with the Wnt/β-catenin and APC mutation status of cancer cells but also closely related to TME eustress. The manuscript is well-written and contributes valuable insights into the role of oxidative stress in CRC. Further in vivo studies and clinical investigations would be beneficial to validate these findings and explore their therapeutic potential.

Limitations of the paper "Differential Regulation of Wingless-WNT/C-JUN N-Terminal 2 Kinase Cross-Talk via Oxidative Eustress in Primary and Metastatic Colorectal Cancer Cells" may include:

The study may have limitations in terms of sample size, potentially impacting the generalizability of the findings to a broader population of CRC patients.

The research appears to be primarily conducted in vitro using cell models (SW480 and SW620 CRC cells). While in vitro studies provide valuable insights, the translation of these findings to in vivo settings and clinical applications may require further validation.

The paper may lack a comprehensive mechanistic understanding of how oxidative eustress precisely modulates the Wnt/β-catenin and JNK signaling pathways in primary and metastatic CRC cells. Further elucidation of the underlying molecular mechanisms could strengthen the study.

The direct clinical relevance of the findings to patient outcomes and treatment strategies for CRC may need to be further explored and validated in clinical trials or patient studies.

Addressing these limitations could enhance the robustness and applicability of the study's findings in the context of colorectal cancer research and therapeutic development.

Additionally,

In the sentence "We demonstrated that metabolic reprogramming in CRC is not only related to the Wnt/ -catenin and APC mutation status of cancer cells but also closely related to TME eustress" on page  14 of the PDF, there is a grammar mistake. The hyphen in "Wnt/ -catenin" should be corrected to "Wnt/β-catenin" for proper representation of the Wnt/β-catenin signaling pathway.

Consider addressing any minor typographical or grammatical errors to improve overall readability.

Overall, this study provides significant contributions to the understanding of oxidative stress and signaling pathways in CRC, with potential implications for therapeutic strategies. Further research in this area is encouraged to build on these promising findings.

Author Response

2° Review Report (Round 1)

These findings shed light on the intricate interplay between oxidative stress, signaling pathways, and metabolic adaptations in CRC cells, offering potential avenues for targeted therapeutic approaches. The study highlights that metabolic reprogramming in CRC is not only associated with the Wnt/β-catenin and APC mutation status of cancer cells but also closely related to TME eustress. The manuscript is well-written and contributes valuable insights into the role of oxidative stress in CRC. Further in vivo studies and clinical investigations would be beneficial to validate these findings and explore their therapeutic potential. 

Limitations of the paper "Differential Regulation of Wingless-WNT/C-JUN N-Terminal 2 Kinase Cross-Talk via Oxidative Eustress in Primary and Metastatic Colorectal Cancer Cells" may include: 

  1. The study may have limitations in terms of sample size, potentially impacting the generalizability of the findings to a broader population of CRC patients.

  1. The research appears to be primarily conducted in vitro using cell models (SW480 and SW620 CRC cells). While in vitro studies provide valuable insights, the translation of these findings to in vivo settings and clinical applications may require further validation.

  1. The paper may lack a comprehensive mechanistic understanding of how oxidative eustress precisely modulates the Wnt/β-catenin and JNK signaling pathways in primary and metastatic CRC cells. Further elucidation of the underlying molecular mechanisms could strengthen the study.

  1. The direct clinical relevance of the findings to patient outcomes and treatment strategies for CRC may need to be further explored and validated in clinical trials or patient studies.

  1. Addressing these limitations could enhance the robustness and applicability of the study's findings in the context of colorectal cancer research and therapeutic development.

Additionally,

  1. In the sentence "We demonstrated that metabolic reprogramming in CRC is not only related to the Wnt/ -catenin and APC mutation status of cancer cells but also closely related to TME eustress" on page 14 of the PDF, there is a grammar mistake. The hyphen in "Wnt/ -catenin" should be corrected to "Wnt/β-catenin" for proper representation of the Wnt/β-catenin signaling pathway.

  1. Consider addressing any minor typographical or grammatical errors to improve overall readability. 

  1. Overall, this study provides significant contributions to the understanding of oxidative stress and signaling pathways in CRC, with potential implications for therapeutic strategies. Further research in this area is encouraged to build on these promising findings.

Autors rebuttal.

We thank the reviewer for his careful reading of the manuscript and his constructive remarks. We have taken the comments on board to improve and clarify the manuscript. Please find below a detailed point-by-point response to all comments (reviewers’ comments in black, our replies in blue).

  1. The cell models we chose to carry out this study are the validated model of colon cancer progression derived from the same patient, SW480 cells from primary CRC and SW620 cells from lymph node metastasis [Hewitt et al., 2000]. These isogenic models present a somatic truncating mutation of the Wnt signaling caretaker APC; they also have an MSS phenotype. These characteristics are detectable in more than 85% of sporadic CRCs.
  2. Translating what is observed in vitro into in vivo settings and clinical applications will require further validation. This will require a considerable economic and technological commitment for the design of prospective studies that can provide information on the characteristics of the microenvironment of the colon, where the primary tumor develops and the metabolism of the "soil", capable of hosting the metastatic "seed". However, several evidences suggested that H2O2 may function as the “fertilizer” in this process, by driving accelerated inflammation, DNA damage and cancer metabolism [Martinez-Outschoorn UE, Lin Z, Trimmer C, Flomenberg N, Wang C, Pavlides S, Pestell RG, Howell A, Sotgia F, Lisanti MP. Cancer cells metabolically "fertilize" the tumor microenvironment with hydrogen peroxide, driving the Warburg effect: implications for PET imaging of human tumors. Cell Cycle. 2011 Aug 1;10(15):2504-20. doi: 10.4161/cc.10.15.16585].

  1. Normal metabolism and steady-state functions require low levels of oxidants, termed “oxidative eustress”. A supraphysiological challenge with oxidants or their inadequate detoxification/inactivation is termed oxidative distress. Oxidative stress is generated endogenously (cell metabolism) and exogenously (exposome). Indeed, many pathophysiological processes have an initiating component of oxidative distress, which opens up the field for research into redox medicine applied mainly to prevention (see Professor Helmut Sies researches). Some observational/mechanistic studies have detected H2O2 production in the epithelium of the normal colon, indicating the involvement of microbial, pollutant, therapeutic and generally xenobiotic products. Currently, the effects of ROS in the TME of primary and metastatic CRC are not fully understood and further studies on the molecular communication of Wnt/β-catenin will be needed.

Although, it has been reported that ERK1/2 signaling phosphorylation appears reduced in early stages of colon tumorigenesis while increased in advanced CRC with metastasis. This regulation of kinase activity in CRC appears to be related to H2O2. Furthermore, it has been suggested that ERK2 may have a growth-promoting role while ERK1 has a regulatory role in colon tumorigenesis. [Parascandolo A, Benincasa G, Corcione F, Laukkanen MO. ERK2 Is a Promoter of Cancer Cell Growth and Migration in Colon Adenocarcinoma. Antioxidants (Basel). 2024]. Under physiological conditions, intracellular H2O2 can be efficiently converted into a hydroxyl radical and lead to an overall increase in 8-oxoG. It plays an important role in cellular metabolic responses but if produced it can induce an increase in DNA lesions in proliferating cells. Oxidative DNA damage is rapidly repaired by the Base Excision Repair system, restoring the redox balance. Furthermore, the elevated level of 8-oxoG per se may not increase cell death [Wang, R.; Li, C.; Qiao, P.; Xue, Y.; Zheng. X.; Chen, H.; Zeng, X.; Liu, W.; Boldogh, I.; Ba, X. OGG1-initiated base excision repair exacerbates oxidative stress-induced parthanatos. Cell Death Dis. 2018, 9, 628].

  1. The direct clinical relevance of the findings to patient outcomes and treatment strategies for CRC may need to be further explored and validated in clinical trials or patient studies. Indeed, the pathophysiology of CRC progression is very complex, future studies will have to take into account not only the tissue micro-environment but also the patient's systemic macro-environment. Translational studies and integrated expertise will therefore be needed to use the patient's biological data for a personalised therapeutic approach to CRC and patient lifestyle care.

  1. Evaluation of the expression of Wnt/b-catenin and JNK signals from complex biological samples could improve the robustness and applicability of the results of this study in the context of colorectal cancer research and therapeutic development.

  1. We changed the incorrect sentence to “We observed that in CRC cells bearing Wnt/b-catenin and APC mutations, metabolic reprogramming may be regulated by oxidative eustress in the TME”.

  1. As suggested by the reviewer, we carefully checked the whole text to make the necessary corrections (shown in the text in blue colour) and properly comprehend the manuscript.

  1. We sincerely thank the reviewer for his appreciation of this work and for encouraging us to continue in this field of study.

Round 2

Reviewer 1 Report

Comments and Suggestions for Authors

The manuscript has been significantly improved, and the authors have explained all the doubts. They also updated the figures as requested. I appreciate that. I have no major comments. Please, accept the paper.

Comments on the Quality of English Language

 Minor editing of English language required

Author Response

We thank the reviewer for his comments.

We have revised the article to improve the English language form.

 We enclose the pdf of the corrected article.
